# Go with the flow:
# Adaptive Control for Neural ODEs

**Mathieu Chalvidal**[1,2,3]**, Matthew Ricci**[4]**, Rufin VanRullen**[1,3]**, Thomas Serre**[1,2]

[1]Artificial and Natural Intelligence Toulouse Institute, Universite de Toulouse, France
[2]Carney Institute for Brain Science, Dpt. of Cognitive Linguistic & Psychological Sciences
Brown University, Providence, RI 02912
[3]Centre de Recherche Cerveau & Cognition CNRS, Université de Toulouse
[4]Data Science Initiative, Brown University, Providence, RI 02912
{mathieu_chalvid, mgr, thomas_serre}@brown.edu
rufin.vanrullen@cnrs.fr

## Abstract

Despite their elegant formulation and lightweight memory cost, neural ordinary differential equations (NODEs) suffer from known representational limitations. In particular, the single flow learned by NODEs cannot express all homeomorphisms from a given data space to itself, and their static weight parameterization restricts the type of functions they can learn compared to discrete architectures with layer-dependent weights. Here, we describe a new module called neurally-controlled ODE (N-CODE) designed to improve the expressivity of NODEs. The parameters of N-CODE modules are dynamic variables governed by a trainable map from initial or current activation state, resulting in forms of open-loop and closed-loop control, respectively. A single module is sufficient for learning a distribution on non-autonomous flows that adaptively drive neural representations. We provide theoretical and empirical evidence that N-CODE circumvents limitations of previous NODEs models and show how increased model expressivity manifests in several supervised and unsupervised learning problems. These favorable empirical results indicate the potential of using data- and activity-dependent plasticity in neural networks across numerous domains.

## 1 Introduction

The interpretation of artificial neural networks as continuous-time dynamical systems has led to both theoretical and practical advances in representation learning. According to this interpretation, the separate layers of a deep neural network are understood to be a discretization of a continuous-time operator so that, in effect, the net is infinitely deep. One important class of continuous-time models, neural ordinary differential equations (NODEs) (Chen et al., 2018), have found natural applications in generative variational inference (Grathwohl et al., 2019) and physical modeling (Köhler et al., 2019; Ruthotto et al., 2020) because of their ability to take advantage of black-box differential equation solvers and correspondence to dynamical systems in nature.

Nevertheless, NODEs suffer from known representational limitations, which researchers have tried to alleviate either by lifting the NODE activation space to higher dimensions or by allowing the transition operator to change in time, making the system non-autonomous (Dupont et al., 2019). For example, Zhang et al. (2020) showed that NODEs can arbitrarily approximate maps from $\mathbb{R}^d$ to $\mathbb{R}$ if NODE dynamics operate with an additional time dimension in $\mathbb{R}^{d+1}$ and the system is affixed with an additional linear layer. The same authors showed that NODEs could approximate homeomorphisms from $\mathbb{R}^d$ to itself if the dynamics were lifted to $\mathbb{R}^{2d}$. Yet, the set of homeomorphisms from $\mathbb{R}^d$ to itself is in fact quite a conservative function space from the perspective of representation learning, since these mappings preserve topological invariants of the data space, preventing them from "disentangling" data classes like those of the annulus data in Fig. 1 (lower left panel). In general, much remains to be understood about the continuous-time framework and its expressive capabilities.

In this paper, we propose a new approach that we call neurally-controlled ODEs (N-CODE) designed to increase the expressivity of continuous-time neural nets by using tools from control theory. Whereas previous continuous-time methods learn a single, time-varying vector field for the whole input space, our system learns a *family* of vector fields parameterized by data. We do so by mapping the input space to a collection of control weights which interact with neural activity to optimally steer model dynamics. The implications of this new formulation are critical for model expressivity.

In particular, the transformation of the input space is no longer constrained to be a homeomorphism, since the flows associated with each datum are specifically adapted to that point. Consequently, our system can easily "tear" apart the two annulus classes in Fig. 1 (lower right panel) without directly lifting the data space to a higher dimension. Moreover, when control weights are allowed to vary in time, they can play the role of fast, plastic synapses which can adapt to dynamic model states and inputs.

The rest of the paper proceeds as follows. First, we will lay out the background for N-CODE and its technical formulation. Then, we will demonstrate its efficacy for supervised and unsupervised learning. In the supervised case, we show how N-CODE can classify data by learning to bifurcate its dynamics along class boundaries as well as memorize high-dimensional patterns in real-time using fast synapses. Then, we show how the flows learned by N-CODE can be used as latent representations in an unsupervised autoencoder, improving image generation over a base model.

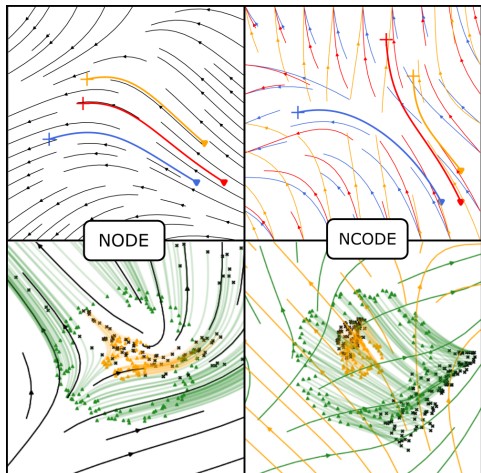

Figure 1: Vector fields for continuous-time neural networks. Integral curves with arrows show the trajectories of data points under the influence of the network. **Top left**: Standard NODEs learn a single time-independent flow (in black) that must account for the whole data space. **Top right**: N-CODE learns a family of vector fields (red vs yellow vs blue), enabling the system to flexibly adjust the trajectory for every data point. **Bottom left**: Trained NODE trajectories of initial values in a data set of concentric annuli, colored green and yellow. The NODE transformation is a homeomorphism on the data space and cannot separate the classes as a result. Colored points are the initial state of the dynamics, and black points are the final state. **Bottom right**: Corresponding flows for N-CODE which easily separate the classes. Transiting from the inner to outer annulus effects a bifurcation which linearly separates the data.

## 2    BACKGROUND

Neural ODEs (NODEs) (Chen et al., 2018) are dynamical systems of the form

$$\frac{d\boldsymbol{x}}{dt} = f(\boldsymbol{x}, \boldsymbol{\theta}, t), \tag{1}$$

where $\mathcal{X}$ is a space of features, $\boldsymbol{\theta} \in \Theta$ is a collection of learnable parameters, and $f : \mathcal{X} \times \Theta \times \mathbb{R} \mapsto \mathcal{X}$ is an equation of motion which we take to be differentiable on its whole domain. $f$ defines a *flow*, i.e. a triple $(\mathcal{X}, \mathbb{R}, \Phi_{\boldsymbol{\theta}})$ with $\Phi_{\boldsymbol{\theta}} : \mathcal{X} \times \mathbb{R} \mapsto \mathcal{X}$ defined by

$$\Phi_{\boldsymbol{\theta}}(\boldsymbol{x}(0), T) = \boldsymbol{x}(0) + \int_0^T f(\boldsymbol{x}(t), \boldsymbol{\theta}, t) dt \tag{2}$$

which relates an initial point $\boldsymbol{x}(0)$ to an orbit of points $\{\boldsymbol{x}(t) = \Phi_\theta(\boldsymbol{x}(0), t), t \in \mathbb{R}\}$. For a fixed $T$, the map $x \mapsto \Phi_{\boldsymbol{\theta}}(x, T)$ is a homeomorphism from $\mathcal{X}$ to itelf parametrized by $\boldsymbol{\theta}$.

Several properties of such flows make them appealing for machine learning. For example, the ODEs that govern such such flows can be solved with off-the shelf solvers and they can potentially model data irregularly sampled in time. Moreover, such flows are reversible maps by construction whose inverse is just the system integrated backward in time, $\Phi_\theta(.,t)^{-1} = \Phi_\theta(.,-t)$. This property enables depth-constant memory cost of training thanks to the adjoint sensitivity method (Pontryagin Lev Semyonovich ; Boltyanskii V G & F, 1962) and the modeling of continuous-time generative normalizing flow algorithms (**?**).

Interestingly, discretizing Eq. 2 yields the recursive formulation of a residual network (He et al., 2015) with a single residual operator $f_{\boldsymbol{\theta}}$:

$$\Phi_{ResNet}(\boldsymbol{x}(0), T) = \boldsymbol{x}(0) + \sum_{t=1}^{T} f_{\boldsymbol{\theta}}(\boldsymbol{x}_{t-1}) \tag{3}$$

In this sense, NODEs with a time-independent (autonomous) equation of motion, $f$, are the infinitely-deep limit of weight-tied residual networks. Relying on the fact that every non-autonomous dynamical system with state $\boldsymbol{x} \in \mathbb{R}^d$ is equivalent to an autonomous system on the extended state $(\boldsymbol{x}, t) \in \mathbb{R}^{d+1}$, Eq. 2 can also be used to model general, weight-untied residual networks. However it remains unclear how dependence of $f$ in time should be modeled in practice and how their dynamics relate to their discrete counterparts with weights evolving freely across blocks through gradient descent.

## 3  N-CODE: LEARNING TO CONTROL DATA-DEPENDENT FLOWS

**General formulation -**  The main idea of N-CODE is to consider the parameters, $\boldsymbol{\theta}(t)$, in Eq. 1 as control variables for the dynamical state, $\boldsymbol{x}(t)$. Model dynamics are then governed by a coupled system of equations on the extended state $\boldsymbol{z}(t) = (\boldsymbol{x}(t), \boldsymbol{\theta}(t))$. The initial value of the control weights, $\boldsymbol{\theta}(0)$, is given by a mapping $\gamma : \mathcal{X} \to \Theta$. Throughout, we assume that the initial time point is $t = 0$. The full trajectory of control weights, $\boldsymbol{\theta}(t)$, is then output by a controller, $g$, given by another differentiable equation of motion $g : \Theta \times \mathcal{X} \times \mathbb{R} \mapsto \Theta$ with initial condition $\gamma(\boldsymbol{x}_0)$. Given an initial point, $\boldsymbol{x}(0)$, we can solve the initial value problem (IVP)

$$\begin{cases} \dfrac{d\boldsymbol{z}}{dt} = h(\boldsymbol{z}, t) \\ \boldsymbol{z}(0) = \boldsymbol{z}_0 \end{cases} = \begin{cases} \left( \dfrac{d\boldsymbol{x}}{dt}, \dfrac{d\boldsymbol{\theta}}{dt} \right) = (f(\boldsymbol{x}, \boldsymbol{\theta}, t), g(\boldsymbol{\theta}, \boldsymbol{x}, t)) \\ (\boldsymbol{x}(0), \boldsymbol{\theta}(0)) = (\boldsymbol{x}_0, \gamma(\boldsymbol{x}_0)) \end{cases}, \tag{4}$$

where $h = (f, g)$. We may think of $g$ and $\gamma$ as a controller of the dynamical system with equation of motion, $f$. We model $g$ and $\gamma$ as neural networks parameterized by $\mu \in \mathbb{R}^{n_\mu}$ and use gradient descent techniques where the gradient can be computed by solving an adjoint problem (Pontryagin Lev Semyonovich ; Boltyanskii V G & F, 1962) that we describe in the next section. Our goal here is to use the meta-parameterization in the space $\Theta$ to create richer dynamic behavior for a given $f$ than directly optimizing fixed weights $\boldsymbol{\theta}$.

**Well-posedness -** If $f$ and $g$ are continuously differentiable with respect to $\boldsymbol{x}$ and $\boldsymbol{\theta}$ and continuous with respect to $t$, then, for all initial conditions $(\boldsymbol{x}(0), \boldsymbol{\theta}(0))$, there exists a unique solution $\boldsymbol{z}$ for Eq. 4 by the Cauchy-Lipschitz theorem. This result leads to the existence and uniqueness of the augmented flow $(\mathcal{X} \times \Theta, \mathbb{R}, \Phi_{\boldsymbol{\mu}})$ with $\Phi_{\boldsymbol{\mu}} : (\mathcal{X} \times \Theta) \times \mathbb{R} \mapsto \mathcal{X} \times \Theta$. Moreover, considering the restriction of such a flow on $\mathcal{X}$, we are now endowed with a universal approximator for at least the set of homeomorphisms on $\mathcal{X}$ given that this restriction constitutes a non-autonomous system. We discuss now how $g$ and $\gamma$ affect the evolution of the variable $\boldsymbol{x}(t)$, exhibiting two forms of control and noting how they relate to previous extensions of NODEs.

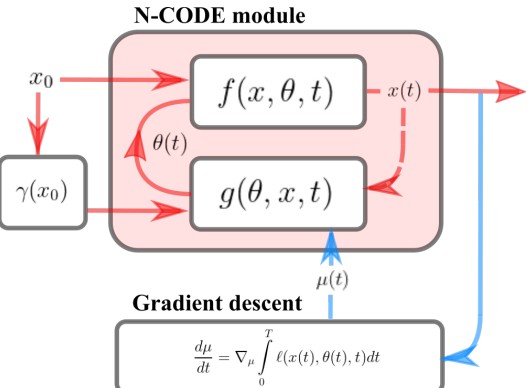

Figure 2: Diagram of a general N-CODE module: Given a initial state $\boldsymbol{x}(0)$, the module consists of an augmented dynamical system that couples activity state $\boldsymbol{x}$ and weights $\boldsymbol{\theta}$ over time (red arrows). A mapping $\gamma$ infers initial control weights $\boldsymbol{\theta}_0$ defining an initial flow (open-loop control). This flow can potentially evolve in time as $\boldsymbol{\theta}$ might be driven by a feedback signal from $\boldsymbol{x}$ (closed-loop, dotted line). This meta-parameterization of $f$ can be trained with gradient descent by solving an augmented adjoint sensitivity system (blue arrows).

## 3.1 OPEN AND CLOSED-LOOP CONTROLLERS

If the controller outputs control weights as a function of the current state, $\boldsymbol{x}(t)$, then we say it is a *closed-loop* controller. Otherwise, it is an *open-loop* controller.

**Open-loop control:** First, we consider the effect of using only the mapping $\gamma$ in Eq. 4 as a controller. Here, $\gamma$ maps the input space $\mathcal{X}$ to $\Theta$ so that $f$ is conditioned on $\boldsymbol{x}(0)$ but not necessarily on $\boldsymbol{x}(t)$ for $t > 0$. In other words, each initial value $\boldsymbol{x}(0)$ evolves according to its own learned flow $(\mathcal{X}, \mathbb{R}, \Phi_{\boldsymbol{\gamma(x(0))}})$. This allows for trajectories to evolve more freely than within a single flow that must account for the whole data distribution and resolves the problem of non-intersecting orbits (see Figure 4). Recently, (Massaroli et al., 2020b) proposed a similar form of data-conditioned open-loop control with extended state $(\boldsymbol{x}(t), \boldsymbol{x}(0))$. This is a version of our method in which $\gamma$ is of the form $\gamma(\boldsymbol{x}) = \theta(0) = [C : id]$ with $C$ a constant vector, $id$ is the identity function, and : denotes concatenation. Our open-loop formulation makes an architectural distinction between controller and dynamics and is consequently generalizable to the following closed-loop formulation.

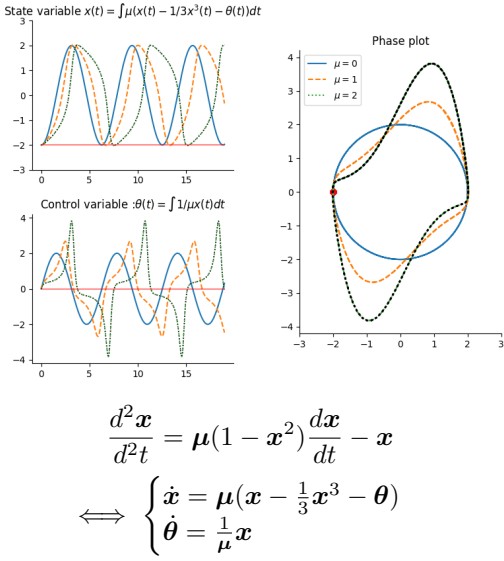

$$\frac{d^2\boldsymbol{x}}{d^2t} = \boldsymbol{\mu}(1 - \boldsymbol{x}^2)\frac{d\boldsymbol{x}}{dt} - \boldsymbol{x}$$

$$\iff \begin{cases} \dot{\boldsymbol{x}} = \boldsymbol{\mu}(\boldsymbol{x} - \frac{1}{3}\boldsymbol{x}^3 - \boldsymbol{\theta}) \\ \dot{\boldsymbol{\theta}} = \frac{1}{\boldsymbol{\mu}}\boldsymbol{x} \end{cases}$$

Figure 3: Example of dynamical control augmentation: The Van der Pol oscillator: As a second order differential equation, the dynamics cannot be approximated by a single 1-dimensional NODE with a constant control $\boldsymbol{\theta}$ (degenerate solution in red). However, if the dynamics are decomposed into a planar system with a dynamic control variable $\boldsymbol{\theta}(t)$, then the parameter $\boldsymbol{\mu}$ can be adapted to fit a particular oscillatory regime (in black). This is a particular form of augmentation discussed in (Dupont et al. (2019); Norcliffe et al. (2020)) that showcases the benefit of using additional variables as evolving parameters of the dynamical system.

**Closed-loop control:** Defining a differentiable mapping, $g$, which outputs the time-dependent control weights $\boldsymbol{\theta}(t)$ given the state of the variable $\boldsymbol{x}(t)$ yields a non-autonomous system on $\mathcal{X}$ (see Fig. 3). This can be seen as a specific dimension augmentation technique where additional variables correspond to the parameters $\boldsymbol{\theta}$. However, contrary to appending extra dimensions which does not change the autonomous property of the system, this augmentation results in a module describing a time-varying transformation $\Phi_{\boldsymbol{\theta(t)}}(\boldsymbol{x_0}, t)$. Note that this formulation generalizes the functional parameterization proposed in recent non-autonomous NODES systems (Choromanski et al., 2020; Massaroli et al., 2020b; **?**), since the evolution of $\boldsymbol{\theta}(t)$ depends on $\boldsymbol{x}(t)$. Much like in the case of classical control theory, we hypothesized that the use of dynamic control weights would be of particular use in reacting to a non-stationary stimulus. We evaluate this hypothesis in section. 5.3.

The expressivity of N-CODE compared to other continuous-time neural networks is encapsulated in the following proposition. The result, proven in appendix, shows that both open-loop and closed-loop control systems overcome NODEs' expressivity constraint with two distinct strategies, data-conditioning and state-space augmentation.

**Proposition 1 -** *There exists a transformation $\phi : \mathbb{R}^d \to \mathbb{R}^d$ which can be expressed by N-CODE but not by NODEs. In particular, $\phi$ is not a homeomorphism.*

## 3.2 TRAINING

**Loss function:** Dynamics are evaluated according to a generalized loss function that integrates a cost over some interval $[0, T]$:

$$l(\boldsymbol{z}) := \int_0^T \ell(\boldsymbol{z}(t), t)dt = \int_0^T \ell(\boldsymbol{x}(t), \boldsymbol{\theta}(t), t)dt \tag{5}$$

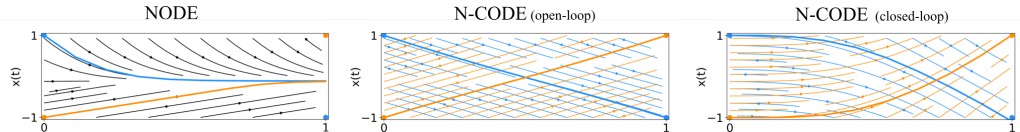

Figure 4: Trajectories over time for three types of continuous-time neural networks learning the 1-dimensional reflection map $\varphi(\boldsymbol{x}) = -\boldsymbol{x}$. **Left:** (*NODE*) Irrespective of the form of $f$, a NODE module cannot learn such a function as trajectories cannot intersect (Dupont et al., 2019). **Middle:** (*Open-loop N-CODE*). The model is able to learn a family of vector fields by controlling a single parameter of $f$ conditioned on $x(0)$. **Right:** (*Closed-loop N-CODE*) With a fixed initialization, $x(0)$, the controller model still learns a deformation of the vector field to learn $\varphi$. Note that the vector field is time-varying in this case, contrary to the two others versions.

The loss in Eq. 5 is more general than in Chen et al. (2018) since $\ell$ can be any Lebesgue-measurable function of both states, $\boldsymbol{x}$, and control parameters, $\boldsymbol{\theta}$. In particular, this includes penalties at discrete time points or over the entire trajectory (Massaroli et al., 2020b) but also regularizations on the weights or activations over the entire trajectory rather than the final state $\boldsymbol{z}(T)$ of the system.

In order to estimate a control function $\boldsymbol{\theta}(t)$ that is optimal with respect to Eq. 5, we invoke Pontryagin's maximum principle (PMP) (Pontryagin Lev Semyonovich ; Boltyanskii V G & F, 1962), which only requires mild assumptions on the functional control space $\Theta$ and applies to functions $f$ that are non-smooth in $\boldsymbol{\theta}$. The PMP gives necessary conditions on $\boldsymbol{\theta}(t)$ at optimality via the augmented adjoint variable $\boldsymbol{a}(t)$. This quantity is the Jacobian of $\ell$ with respect to both $\boldsymbol{x}(t)$ and $\boldsymbol{\theta}(t)$. In the case of $\boldsymbol{\theta}(t)$ being differentiable with respect to the meta-parameters $\boldsymbol{\mu}$, solving for the augmented adjoint state $\boldsymbol{a}(t)$ as in Chen et al. (2018) allows us to compute the gradient of the loss with respect to $\boldsymbol{\mu}$ thanks to Theorem 1.

**Theorem 1 - Augmented adjoint method:** Given the IVP of equation 4 and for $\ell$ defined in equation 5, we have:

$$\frac{\partial l}{\partial \boldsymbol{\mu}} = \int_0^T \boldsymbol{a}(t)^T \frac{\partial h}{\partial \boldsymbol{\mu}} dt, \ \text{ such that } \boldsymbol{a} \text{ satisfies } \begin{cases} \dfrac{d\boldsymbol{a}}{dt} = -\boldsymbol{a}^T \cdot \dfrac{\partial h}{\partial \boldsymbol{z}} - \dfrac{\partial \ell}{\partial \boldsymbol{z}} \\ \boldsymbol{a}(T) = \boldsymbol{0} \end{cases} \tag{6}$$

where $\dfrac{\partial h}{\partial \boldsymbol{z}}$ is the Jacobian of $h$ with respect to $\boldsymbol{z}$: $\quad \dfrac{\partial h}{\partial \boldsymbol{z}} = \begin{pmatrix} \frac{\partial f}{\partial \boldsymbol{x}} & \frac{\partial f}{\partial \boldsymbol{\theta}} \\ \frac{\partial g}{\partial \boldsymbol{x}} & \frac{\partial g}{\partial \boldsymbol{\theta}} \end{pmatrix}.$

In practice, we compute the Jacobian for this augmented dynamics with open source automatic differentiation libraries using Pytorch (Paszke et al., 2019), enabling seamless integration of N-CODE modules in bigger architectures. We show an example of such modules in section B of Appendix.

## 4 EXPERIMENTS

### 4.1 REFLECTION AND CONCENTRIC ANNULI

We introduce our approach with the 1- and 2-dimensional problems discussed in (Dupont et al., 2019; Massaroli et al., 2020b), consisting of learning either the reflection map $\varphi(\boldsymbol{x}) = -\boldsymbol{x}$ or a linear classification boundary on a data-set of concentric annuli. Earlier results have shown that vanilla NODEs cannot learn these functions since NODEs preserve the topology of the data space, notably its linking number, leading to unstable and complex flows for entangled classes. However, both the open and closed loop formulations of N-CODE easily fit these functions, as shown in Figs 4 and 5. The adaptive parameterization allows N-CODE to learn separate vector fields for each input, allowing for simpler paths that ease model convergence. Informally, we may think of the classification decisions as being mediated by a bifurcation parameterized by the data space. Transiting across the true classification boundary switches the vector field from one pushing points to one side of the classification boundary to the other. Moreover, the construction of $\gamma$ and $g$ as a neural network guarantees a smooth variation of the flow with respect to the system state, which can potentially provide interesting regularization properties on the family learned by the model.

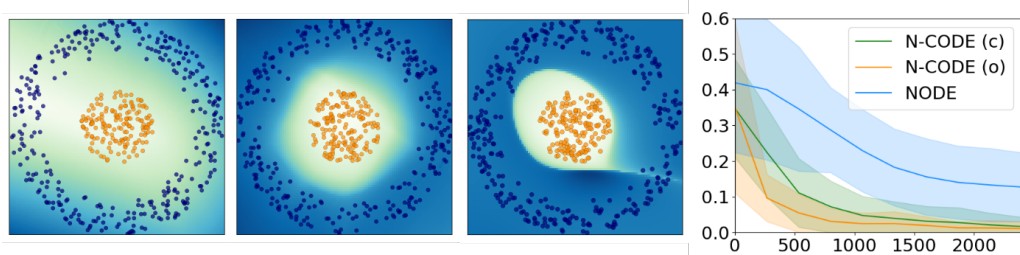

Figure 5: The *concentric annuli* problem. A classifier must separate two classes in $\mathbb{R}^2$, a central disk and an annulus that encircles it. **Left**: Soft decision boundaries (blue to white) for NODE (*First*), N-CODE open-loop (*Second*) and closed-loop (*First*) models. **Right**: Training curve for the three models.

## 4.2 SUPERVISED IMAGE CLASSIFICATION

We now explore different data-dependent controls for continuous-time models on an image classification task. We tested the open-loop version of N-CODE against augmented NODEs (Dupont et al., 2019) and the data-control model of (Massaroli et al., 2020b). We perform this experiment on the MNIST and CIFAR-10 image datasets. Our model consists of the dynamical system with equation of motion $f$ expressed as a single convolutional block $(1 \times 1 \mapsto 3 \times 3 \mapsto 1 \times 1)$ with 50 channels. The control weights $\boldsymbol{\theta} \in \mathbb{R}^K$ are a real-valued vector of $K$ concatenated parameters instantiating the convolution kernels.

| Model | MNIST | CIFAR-10 |
|---|---|---|
| ANODE | 98.2% | 61.2% |
| NODE DC | 98.5% | 62.0% |
| N-CODE (open) | **99.2%** | **74.1%** |

Table 1: Classification accuracy on MNIST and CIFAR10 test sets for NODE, NODE with data-dependant flow (NODE DC) and open-loop N-CODE (ours)

The 10-class prediction vector is obtained as a linear transformation of the final state $\boldsymbol{x}(T)$ of the dynamics with a softmax activation. We aimed at simplicity by defining the mapping $\gamma$ as a two-layer perceptron with 10 hidden units. We varied how $\gamma$ affects $\boldsymbol{\theta}$ by defining different linear heads for each convolution in the block. (see C.2 of Appendix). We did not perform any fine-tuning, although we did augment the state dimension, akin to ANODE, by increasing the channel dimension by 10, which helped stabilize training. We show that the open-loop control outperforms competing models. Results are shown in Table. 1 with further details in Section C.2 of Appendix.

## 4.3 REAL-TIME PATTERN MEMORIZATION

To validate our intuition that closed-loop control is particularly valuable on problems involving flexible adaptation to non-stationary stimuli and to distinguish our system from related data-conditioned methods, we trained N-CODE on a dynamic, few-shot memorization task (Miconi et al., 2018). Here, the model is cued to quickly memorize sets of sequentially presented $n$-dimensional binary patterns and to reconstruct one of these patterns when exposed to a degraded version. For each presentation episode, we exposed the model sequentially to $m = 2, 5,$ or $10$ randomly generated $n$-length ($n = 100, 1000$) bit patterns with $\{-1, 1\}$-valued bits. Then, we presented a degraded version of one of the previous patterns in which half of the bits were zeroed, and we tasked the system with reconstructing this degraded pattern via the dynamic state, $\boldsymbol{x}(t)$. This is a difficult task since it requires the memorization of several high-dimensional data in real-time. Miconi et al. (2018) have shown that, although recurrent networks are theoretically able to solve this task, they struggle during learning, while models storing transient information in dynamic synapses tend to perform better. Following this result, we evaluated the performance of a closed-loop N-CODE model in which the output of the control function takes the form of dynamic synapses (Eq. 7). Here, the control parameters are a matrix of dynamic weights, $\boldsymbol{\theta}(t) \in \mathbb{R}^{m \times m}$, which are governed by the outer product $\mu \odot \boldsymbol{x}\boldsymbol{x}^T$,

$$\begin{cases} \left( \dfrac{d\boldsymbol{x}}{dt}, \dfrac{d\boldsymbol{\theta}}{dt} \right) = \left( \rho\left(\boldsymbol{\theta}\boldsymbol{x}\right), \mu \odot \boldsymbol{x}\boldsymbol{x}^T \right) \\ (\boldsymbol{x}(0), \boldsymbol{\theta}(0)) = (\boldsymbol{x}_{\text{stim}}, \boldsymbol{\theta}_0) \end{cases} \tag{7}$$

where $\boldsymbol{\mu}$ is matrix of learned parameters, $\boldsymbol{\theta}_0$ is a learned initial condition, $\rho$ is an element-wise sigmoidal non-linearity, $\odot$ is element-wise multiplication, and $x_{\text{stim}}$ is the presented stimulus. We

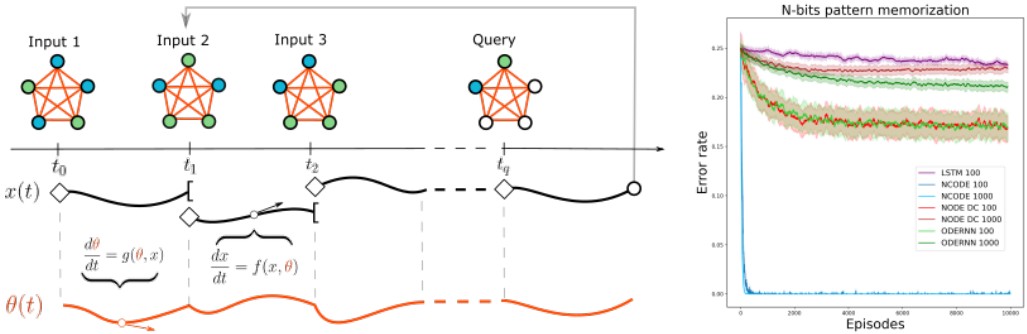

Figure 6: **Left**: Diagram of an episode of the few-shot memorization task Miconi et al. (2018). Three, 5-length $-1$, 1-valued bit patterns ($n = 5$ here for visualization) are presented to the system at regular intervals. At query time, $t_q$, a degraded version of one of the patterns is presented and the task of the model is to complete this pattern. **Right**: Performance of a closed-loop N-CODE, an LSTM, a continuous-time ODE-RNN Rubanova et al. (2019) and the data-conditioned NODE of Massaroli et al. (2020b) averaged over 3 runs. All models besides N-CODE plateau at high values (chance is .25). N-CODE, on the other hand, learns immediately and with extremely high accuracy across all $n$. (see Appendix)

have suppressed $t$ for concision. We ran this experiment on the closed-loop version of N-CODE as well as an LSTM, continuous-time ODE-RNN (Rubanova et al., 2019) and the data-controlled model of Massaroli et al. (2020b). All systems had the same number of learnable parameters, except for the LSTM which was much larger. We found that all considered models, except closed-loop N-CODE, struggled to learn this task (Fig. 6). Note that chance is .25 since half the bits are degraded. A discrete LSTM endowed with a much larger hidden state of 5000 neurons learns marginally better than chance level. ODE-RNN and data control versions plateau at 17.5% error rate. For both $n$ tested, N-CODE learned the problem not only much better, but strikingly faster.

## 4.4 UNSUPERVISED LEARNING : IMAGE AUTOENCODING WITH CONTROLLED FLOW

Finally, we use our control approach in an autoencoding model for low-dimensional representation learning in order to see if merely adding N-CODE to a simpler model can improve performance. Our idea is to use the solution to a system of linear ODEs to encode the model's latent representations. First, images from $\mathcal{U} \in \mathbb{R}^n$ are mapped by an encoder to a space of couplings, $\boldsymbol{\theta}$, by

$$\gamma_{\boldsymbol{\mu}} : \boldsymbol{u} \to \gamma_{\boldsymbol{\mu}}(\boldsymbol{u}) = \boldsymbol{\theta} \in \mathbb{R}^{m \times m}. \tag{8}$$

The latent code is taken as the vector $\boldsymbol{x}(T)$ acquired by solving the autonomous differential system presented in Eq. 9 from 0 to $T$ starting from the initial condition $(\boldsymbol{x}_0, \gamma_{\boldsymbol{\mu}}(u))$ with $\boldsymbol{x}_0 \sim \mathcal{N}(0, I)$:

$$\begin{cases} \left( \dfrac{d\boldsymbol{x}}{dt}, \dfrac{d\boldsymbol{\theta}}{dt} \right) = (\boldsymbol{\theta}\boldsymbol{x}, 0) \\ (\boldsymbol{x}(0), \boldsymbol{\theta}(0)) = (\boldsymbol{x}_0, \gamma_{\boldsymbol{\mu}}(\boldsymbol{u})) \end{cases} \tag{9}$$

This model, which we call autoN-CODE, can be interpreted as a hybrid version of autoencoders and generative normalizing flows, where the latent generative flow is data-dependent and parameterized by the encoder output.

We found that a simple linear system was already more expressive than a discrete linear layer in shaping the latent representation. Since the output dimension of the encoder grows quadratically with the desired latent code dimension of the bottleneck, we adopted a sparse prediction strategy, where as few as two elements of each row of $\boldsymbol{\theta}$ were non-zero, making our model match exactly the number of parameters of a VAE with same architecture. We believe that the increase in representational power over a vanilla autoencoder, as shown by improvements in Fréchet Inception Distance (FID; Heusel et al. (2017)) and more disentangled latent representations (see A.7), despite these simplifying assumptions, shows the potential of this approach for representation learning (Fig. 2). Training details can be found in Sec. C.4.

Here we measure the marginal improvement acquired by adding latent flows to a baseline VAE. In order to endow our deterministic model with the ability to generate image samples, we perform,

similarly to (Ghosh et al., 2020), a post hoc density estimation of the distribution of the latent trajectories' final state. Namely, we employ a gaussian mixture model fit by expectation-maximization and explore the effect of increasing the number of components in the mixture. We report in Table 2, contrary to Ghosh et al. (2020), a substantial decrease in FID of our sampled images to the image test set with an increasing number of components in the mixture (see Fig. 7 for latent evolution and Fig. 8 for sampled examples).

### 4.5 IMAGE GENERATION

Considering the identical architecture of the encoder and decoder to a vanilla model, these results suggest that the optimal encoding control formulation produced a structural change in the latent manifold organization. (See section C.7 in Appendix). Finally, we note that

| FID ($\downarrow$) | CIFAR-10 | CelebA-(64) |
|---|---|---|
| VAE[†](Kingma & Welling, 2013) | 105.45 | 68.07 |
| Auto-encoder[†] (100 components) | 73.24 | 63.11 |
| Auto-encoder[†] (1000 components) | 55.55 | 55.60 |
| AutoNCODE (100 components) | 34.90 | 55.27 |
| AutoNCODE (1000 components) | **24.19** | **52.47** |

Table 2: Frechet Inception Distance (FID) for several recent architectures (lower is better). "Components" are the numbers of components used in the gaussian mixture estimation of the latent distribution.

this model offers several original possibilities to explore, such as sampling from the control coefficients, or adding a regularization on the dynamic evolution as in (Finlay et al., 2020), which we leave for future work.

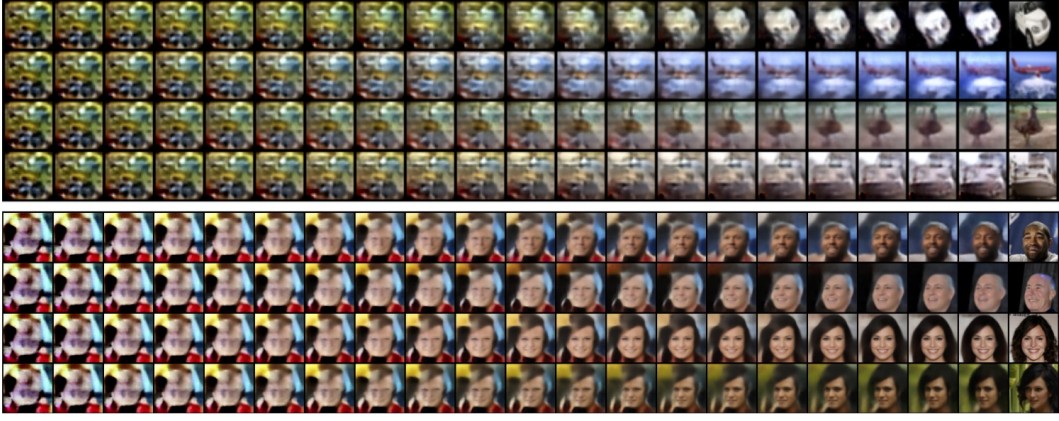

Figure 7: Left to right: Temporal evolutions of an image reconstruction along controlled *orbits* of **AutoNCODE** for representative CIFAR-10 and CelabA test images. Last column: ground truth image.

## 5 RELATED WORK

**Neural ODEs -** This work builds on the recent development of Neural ODEs, which have shown great promise for building generative normalizing flows (Grathwohl et al., 2019) used for modeling molecular interactions (Köhler et al., 2019) or in mean field games (Ruthotto et al., 2019). More recent work has focused on the topological properties of these models (Dupont et al., 2019; Zhang et al., 2020), introducing time-dependent parameterization (Zhang et al., 2019b; Massaroli et al., 2020b; Choromanski et al., 2020), developing novel regularization methods using optimal transport or stochastic perturbations (Finlay et al., 2020; Oganesyan et al., 2020) and adapting them to stochastic differential equations (Tzen & Raginsky, 2019).

**Optimal Control -** Several authors have interpreted deep learning optimization as an optimal control problem (Li et al., 2017; Benning et al., 2019; Liu & Theodorou, 2019; Zhang et al., 2019a; Seidman et al., 2020) providing strong error control and yielding convergence results for alternative algorithms to standard gradient-based learning methods. Of particular interest is the *mean-field optimal control* formulation of (E et al., 2018), which notes the dependence of a unique control variable governing the network dynamics on the whole data population.

**Hypernetworks -** Finally, our system is related to network architectures with adaptive weights such as hypernetworks (Ha et al., 2017), dynamic filters (Jia et al., 2016) and spatial transformers (Jaderberg et al., 2015). Though not explicitly formulated as neural optimal control, these approaches effectively implement a form of modulation of neural network activity as a function of input data and activation state, resulting in compact and expressive models. These methods demonstrate, in our opinion, the significant untapped potential value in developing dynamically controlled modules for deep learning.

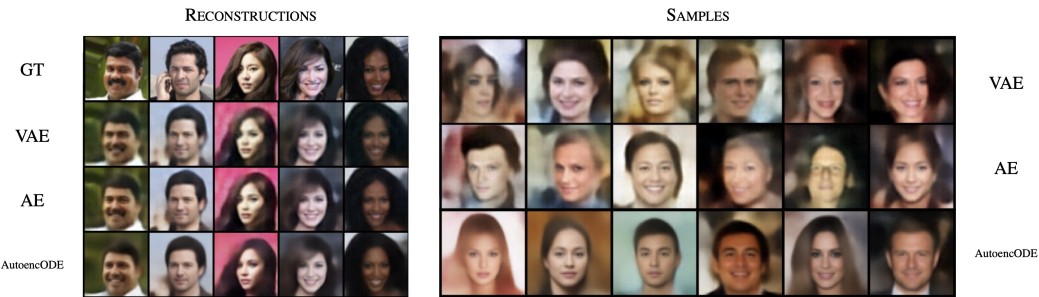

Figure 8: **Left** Reconstructions of random test images from CelebA for different autoencoding models. **Right**: Random samples from the latent space. Deterministic models are fit with 100-components gaussian mixture model.

## 6 CONCLUSION

In this work, we have presented an original control formulation for continuous-time neural feature transformations. We have shown that it is possible to dynamically shape the trajectories of the transformation module applied to the data by augmenting the network with a trained control mechanism, and further demonstrated that this can be applied in the context of supervised and unsupervised representation learning. In future work, we would like to investigate the robustness and generalization properties of such controlled models as well as their similarities with fast-synaptic modulation systems observed in neuroscience, and test this on natural applications such as recurrent neural networks and robotics. An additional avenue for further research is the connection between our system and the theory of bifurcations in dynamical systems and neuroscience.

## ACKNOWLEDGMENTS AND DISCLOSURE OF FUNDING

This work was funded by the ANR-3IA Artificial and Natural Intelligence Toulouse Institute (ANR-19-PI3A-0004).

Additional support provided by ONR (N00014-19-1-2029), NSF (IIS-1912280, OSCI-DEEP ANR grant (ANR-19-NEUC-0004) and the Center for Computation and Visualization (CCV). We acknowledge the Cloud TPU hardware resources that Google made available via the TensorFlow Research Cloud (TFRC) program as well as computing hardware supported by NIH Office of the Director (S10OD025181).

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

APPENDIX

## A  THEORETICAL RESULTS

### A.1  PROOF OF PROPOSITION 1

In order to show that the set of functions defined by N-CODE are not necessarily homeomorphisms on $\mathcal{X}$, we show that they are not generally injective.

**Open-loop**: Consider the one-dimensional N-CODE system $\Phi : (\mathcal{X}, \mathbb{R}) \mapsto \mathcal{X}$ with equation of motion $f(\boldsymbol{x}, \boldsymbol{\theta}) = -\theta$ where $\theta = \gamma(\boldsymbol{x}(0)) = \boldsymbol{x}(0)$. This system has solution

$$\Phi(\boldsymbol{x}(0), T) = \boldsymbol{x}(0) + \int f(\boldsymbol{x}, \boldsymbol{\theta}, t)\, dt$$
$$= \boldsymbol{x}(0) - \boldsymbol{x}(0)T. \tag{10}$$

However, this implies

$$\Phi(0, 1) = \Phi(1, 1) = 0, \tag{11}$$

giving the result.

**Closed-loop**: Similarly, we can show that the following 2D oscillatory system:

$$\begin{cases} \dot{\boldsymbol{x}}(t) = \boldsymbol{\theta}(t) \\ \dot{\boldsymbol{\theta}}(t) = -\boldsymbol{x}(t) \end{cases} \implies \boldsymbol{x}(t) = \alpha\cos(t) + \beta\sin(t) \implies \boldsymbol{x}(t) = \boldsymbol{x}(0)\cos(t) \ \text{if} \ \boldsymbol{\theta}(0) = 0 \tag{12}$$

But $\Phi(x, \pi/2) = 0$, so $\boldsymbol{x} \mapsto \Phi(\boldsymbol{x}, t) = \boldsymbol{x}(t)$ is not injective.

This shows that the two control forms, open and closed, are not restricted to learn homemorphisms on the space $\mathcal{X}$.

### A.2  PROOF OF THEOREM 1

*Proof.* The proof is inspired from Massaroli et al. (2020a). We place our analysis in the Banach space $\mathbb{R}^n$. Since $\ell$ and $h$ are continuously differentiable, let us form the *Lagrangian* functional $\mathcal{L}$ where $\boldsymbol{a}$ is an element of the dual space of $\boldsymbol{z} = \begin{bmatrix} \boldsymbol{x} \\ \boldsymbol{\theta} \end{bmatrix}$:

$$\mathcal{L}(\boldsymbol{z}, \boldsymbol{a}, t) := \int_0^T [\ell - \langle \boldsymbol{a}, \frac{\partial \boldsymbol{z}}{\partial t} - h \rangle] dt \tag{13}$$

The constraint $\langle \boldsymbol{a}(t), \frac{\partial \boldsymbol{z}}{\partial t} - h(\boldsymbol{z}, t) \rangle$ is always *active* on the admissible set by construction of $\boldsymbol{z}$ in equation 4, such that we have $\forall \boldsymbol{z}, \boldsymbol{a}, \quad \frac{\partial \mathcal{L}}{\partial \boldsymbol{\mu}}(\boldsymbol{z}, \boldsymbol{a}, t) = \frac{\partial l}{\partial \boldsymbol{\mu}}(\boldsymbol{z}, \boldsymbol{a}, t)$. Moreover, integrating the left part of the integral in equation 13 gives:

$$\int_0^T \langle \boldsymbol{a}, \frac{\partial \boldsymbol{z}}{\partial t} \rangle dt = \langle \boldsymbol{a}, \boldsymbol{z} \rangle \big|_0^T - \int_0^T \langle \frac{\partial \boldsymbol{a}}{\partial t}, \boldsymbol{z} \rangle dt \tag{14}$$

Now, given that $\ell - \langle \frac{d\boldsymbol{a}}{dt}, \boldsymbol{z} \rangle$ is differentiable in $\mu$ for any $t \in [0, T]$, Leibniz integral rule allows to write

$$\frac{\partial l}{\partial \boldsymbol{\mu}} = \int_0^T \frac{\partial}{\partial \boldsymbol{\mu}}[\ell + \langle \frac{d\boldsymbol{a}}{dt}, \boldsymbol{z} \rangle + \langle \boldsymbol{a}, h \rangle] dt + \langle \boldsymbol{a}(T), \frac{\partial \boldsymbol{z}(T)}{\partial \boldsymbol{\mu}} \rangle - \langle \boldsymbol{a}(0), \frac{\partial \boldsymbol{z}(0)}{\partial \boldsymbol{\mu}} \rangle \tag{15}$$

$$\frac{\partial l}{\partial \boldsymbol{\mu}} = \int_0^T [\frac{\partial \ell}{\partial \boldsymbol{z}} \frac{\partial \boldsymbol{z}}{\partial \boldsymbol{\mu}} + \langle \frac{d\boldsymbol{a}}{dt}, \frac{\partial \boldsymbol{z}}{\partial \boldsymbol{\mu}} \rangle + \langle \boldsymbol{a}, \frac{\partial h}{\partial \boldsymbol{\mu}} \rangle] dt + \langle \boldsymbol{a}(T), \frac{\partial \boldsymbol{z}(T)}{\partial \boldsymbol{\mu}} \rangle \tag{16}$$

$$\frac{\partial l}{\partial \boldsymbol{\mu}} = \int_0^T [\frac{\partial \ell}{\partial \boldsymbol{z}} \frac{\partial \boldsymbol{z}}{\partial \boldsymbol{\mu}} + \langle \frac{d\boldsymbol{a}}{dt}, \frac{\partial \boldsymbol{z}}{\partial \boldsymbol{\mu}} \rangle + \langle \boldsymbol{a}, \frac{\partial h}{\partial \boldsymbol{\mu}} + \frac{\partial h}{\partial t} \frac{\partial t}{\partial \boldsymbol{\mu}} + \frac{\partial h}{\partial \boldsymbol{z}} \frac{\partial \boldsymbol{z}}{\partial \boldsymbol{\mu}} ] dt + \langle \boldsymbol{a}(T), \frac{\partial \boldsymbol{z}(T)}{\partial \boldsymbol{\mu}} \rangle \tag{17}$$

The last equation can be reordered as:

$$\frac{\partial l}{\partial \boldsymbol{\mu}} = \int_0^T \langle \frac{d\boldsymbol{a}}{dt} - \boldsymbol{a}\frac{\partial h}{\partial \boldsymbol{z}} - \frac{\partial \ell}{\partial \boldsymbol{z}}, \frac{\partial \boldsymbol{z}}{\partial \boldsymbol{\mu}} \rangle dt + \int_0^T \langle \boldsymbol{a}, \frac{\partial h}{\partial \boldsymbol{\mu}} \rangle dt + \langle \boldsymbol{a}(T), \frac{\partial \boldsymbol{z}(T)}{\partial \boldsymbol{\mu}} \rangle \tag{18}$$

Posing that $\boldsymbol{a}(T) = \mathbb{O}_{|\mathcal{X}|+|\Theta|}$ and $\langle \frac{d\boldsymbol{a}}{dt} - \boldsymbol{a}\frac{\partial h}{\partial \boldsymbol{z}} - \frac{\partial \ell}{\partial \boldsymbol{z}}, \frac{\partial \boldsymbol{z}}{\partial \boldsymbol{\mu}} \rangle = 0$, the result follows. $\qquad\square$

**Remark 1:** This result allows to compute gradient estimate assuming that $\boldsymbol{\mu}$ directly parametrizes $h$ such that the gradient $\frac{\partial h}{\partial \boldsymbol{\mu}}$ are straightforward to compute. However, in the case of optimizing the initial mapping $\gamma$ that infer the variable $\boldsymbol{\theta}$, this result can be combined with usual chain rule to estimate the gradient:

$$\frac{\partial l}{\partial \boldsymbol{\mu}_\gamma} = \frac{\partial \theta}{\partial \gamma} \frac{\partial l}{\partial \boldsymbol{\theta}} \tag{19}$$

## B  IMPLEMENTATION

**Generic N-CODE module -** We provide here a commented generic PyTorch implementation for the N-CODE module in the open-loop setting.

```python
class NCODE_func(torch.nn.Module):
    def __init__(self, *args):
        super(NCODE_func, self).__init__()

        #Definition of the ODE system
        self.f = ...
        self.g = ...

    def forward(self, t, z):

        #Unpack variables (theta_t is potentially a tuple of tensors)
        x_t, *theta_t = z

        #Compute evolution od the system
        d_x_t = self.f(x_t,theta_t)
        d_theta_t = self.g(theta_t,d_x_t = f(x_t,theta_t)

        return (d_x_t, *d_theta_t)

class NCODE_block(torch.nn.Module):
    def __init__(self, *args):
        super(NCODE_block, self).__init__()

        #Definition of module and controller
        self.gamma = ...
        self.NCODE_func = ...

    def forward(self, x_0):

        #Set initial conditions
        self.theta_0 = self.gamma(x_0)
```

```
    z_0 = (x_0,*self.theta_0)

    #Run the system forward with defined settings
    x_t, *theta_t = odeint_adjoint(self.odefunc, z_0, integration_time, rtol, at

    return (x_t, *theta_t)
```

## C  EXPERIMENTAL RESULTS

**Resources -** Our experiments were run on a 12GB NVIDIA® Titan Xp GPUs cluster equipped with CUDA 10.1 driver. Neural ODEs were trained using the `torchdiffeq` (Chen et al., 2018) PyTorch package.

### C.1  FLOWS FOR ANNULI DATASET

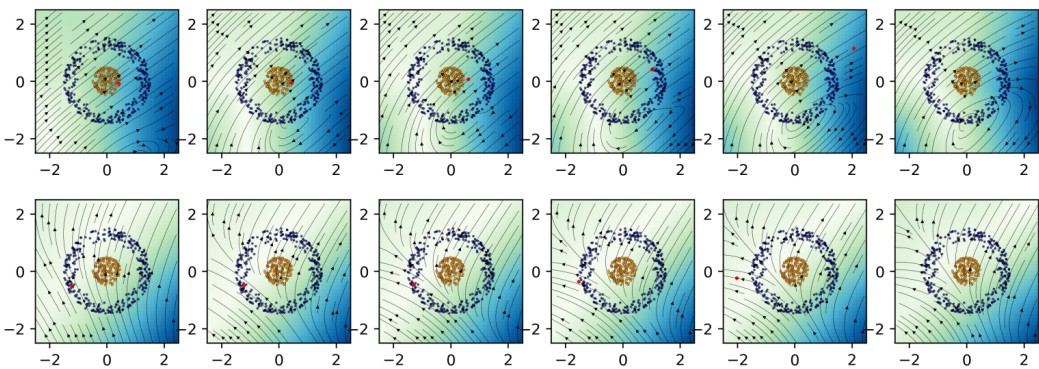

Figure A.1: Time-varying flows for two points in different classes (inner: top row; outer: bottom row) of the annuli dataset. Time advances from left to right.

### C.2  SUPERVISED IMAGE CLASSIFICATION

For MNIST and CIFAR-10 classification, we defined the equation of motion $f$ as sequence of convolutional filters replicating a ResNet block architecture with padding to conserve dimensionality and non-linear ReLU activation. The base architecture takes one of the forms

- $1 \times 1$, $k$ filters, 1 stride, 0 padding
- $3 \times 3$, $k$ filters, 1 stride, 1 padding
- $1 \times 1$, $c + a$ filters, 1 stride, 0 padding

with $k$ the number of filters, $c$ the number of channels (1 for MNIST and 3 for CIFAR10) and $a$ the augmentation channel (Dupont et al., 2019). The open-loop controller is a very simple linear transformation of the data into 10 hidden units followed by multiple heads outputting the vector of weights for each convolution kernel. In order to examine the effect of the size of dimensionality of the equation of motion on learnability, we tried different parameterizations for $f$:

- *NCODE* : The three kernels are conditioned by the controller $\gamma$.
- *NCODE* $3 \times 3$: The inner 3x3 kernels are dynamically conditioned.
- *NODE DC* : This corresponds to the form of data-control proposed by (Massaroli et al., 2020b) where the image data is concatenated with state $x(t)$ at each evaluation of the first 1x1 convolution kernel.

All parameters not adaptively controlled are simply learned as fixed parameters. For all models we used the adaptative solver Dormund-Prince with tolerance of 1e-3.

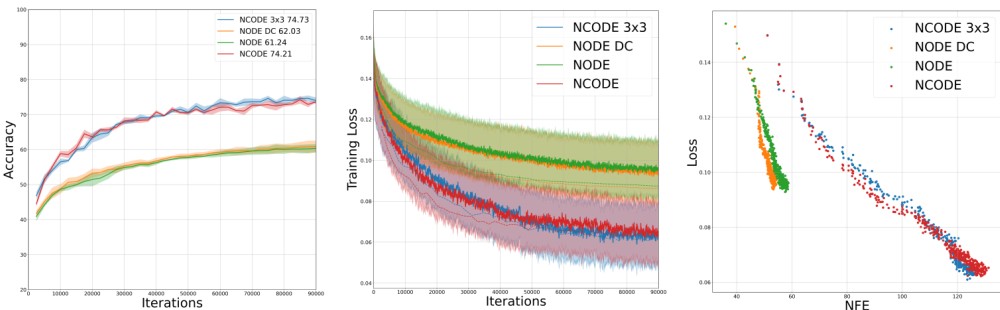

Figure A.2: Accuracy, Train and test losses over training interations and Number of function evaluations (NFE) vs loss for several versions of NCODE on CIFAR-10.

The results show that N-CODE models achieve higher accuracy in fewer epochs and overall lower loss (Fig. A.2, top panels). However, the simplicity of the mapping $\gamma$ in this experiment and the absence of regularization of the induced map $\Phi_\gamma$ (Finlay et al., 2020) yields a complex flow requiring a higher total number of function evaluations (Fig. A.2, lower panels). This potentially prevents NCODE from achieving even better accuracy. Interestingly, the results for the partially controlled systems suggest that gradient descent accommodates the learning of hybrid modules $f$ with both static and dynamic convolution parameters. These results advocate for the exploration of specific closed-loop formulations and regularization of control for convolution layers which we leave for future work.

### C.3  PATTERN MEMORIZATION

This experiment is an update of Miconi et al. (2018), which showed that using a simple Hebbian update rule between weights endowed recurrent networks with fast memory properties. The base model is a single-cell recurrent network with all-to-all coupling, $\boldsymbol{\theta}(t)$. The considered models are

*Open-loop control*: We adopt the implementation of (Massaroli et al., 2020b) by appending the presented pattern $\boldsymbol{x}_{stim}$ to the dynamic variable $\boldsymbol{x}(t)$ such that $\boldsymbol{\theta}(t) = \boldsymbol{\theta}$ is linear layer $L(\mathbb{R}^{2 \times N}, \mathbb{R}^N)$.

*Closed-loop control*: The weights are fully dynamic and the influence of the plastic evolution is tuned by learning the components $\mu$ that apply an element-wise gain for every weight velocity, according to Eq. 7.

*ODE-RNN*: Here, a static-weight NODE models the evolution of a continuous-time single-cell recurrent network whose hidden state is carried over from one presentation to the next. A supplementary linear read-out $L(\mathbb{R}^N, \mathbb{R}^N)$ outputs the model's guess and balances the number of parameters with other models.

*LSTM*: A vanilla LSTM cell (Hochreiter & Schmidhuber, 1997). In coherence with (Miconi et al., 2018) results, the hidden state dimension need to be greatly increased for the module to start memorizing. We tested a 5000-dimensional hidden state for the reported results.

For each model, we learn the model weights with gradient descent using an Adam optimizer with a learning rate $\lambda = 3e - 4$ and use the $L_2$ distance between the reconstruction and objective pattern as our objective function. For continuous models, each episode consists of a sequential presentation of each pattern for 0.5 sec followed by a query time of 0.5 sec. For the LSTM, we adopted the setting of (Miconi et al., 2018) where the sequence of presentation is discretized into 5 time-steps of presentations. We tested for episodes of 3,5 and 10 patterns with 2 presentations in random order and different proportions of degradation (0.5,0.7 and 0.9). Static-weight model performance rapidly deteriorated in more challenging settings, whereas N-CODE reconstruction converged to a residual error below 1% in all cases.

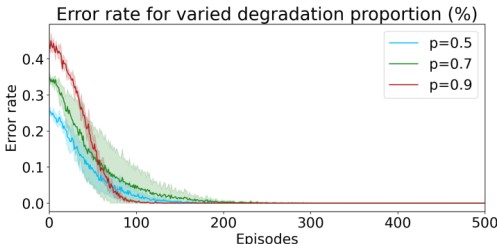 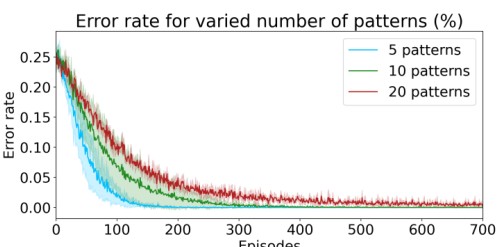

Figure A.3: Error rate of N-CODE on the 1000-bit reconstruction task as a function of episodes for (**Left**) different proportions of degraded bits and (**Right**) different number of patterns presented in one episode. In all cases, the model learns in several hundreds episodes.

|  | **MNIST** | **CIFAR-10** | **CelebA** |
|---|---|---|---|
|  | $\mathbf{x} \in \mathbb{R}^{3 \times 28 \times 28}$ | $\mathbf{x} \in \mathbb{R}^{3 \times 32 \times 32}$ | $\mathbf{x} \in \mathbb{R}^{3 \times 64 \times 64}$ |
| **Encoder**: | Flatten | $\text{Conv}_{256} \mapsto \text{BN} \mapsto \text{Relu}$ | $\text{Conv}_{256} \mapsto \text{BN} \mapsto \text{Relu}$ |
|  | $\text{FC}_{400} \mapsto \text{Relu}$ | $\text{Conv}_{512} \mapsto \text{BN} \mapsto \text{Relu}$ | $\text{Conv}_{512} \mapsto \text{BN} \mapsto \text{Relu}$ |
|  | $\text{FC}_{25} \mapsto \text{Norm}$ | $\text{Conv}_{1024} \mapsto \text{BN} \mapsto \text{Relu}$ | $\text{Conv}_{1024} \mapsto \text{BN} \mapsto \text{Relu}$ |
|  |  | $\text{FC}_{128*2} \mapsto \text{Norm}$ | $\text{FC}_{64*10} \mapsto \text{Norm}$ |
| **Latent dynamics**: | ODE[0,10] | ODE[0,10] | ODE[0,10] |
| **Decoder**: | $\text{FC}_{400} \mapsto \text{Relu}$ | $\text{FC}_{1024*8*8} \mapsto \text{Norm}$ | $\text{FC}_{1024*8*8} \mapsto \text{Norm}$ |
|  | $\text{FC}_{784} \mapsto \text{Sigmoid}$ | $\text{ConvT}_{512} \mapsto \text{BN} \mapsto \text{Relu}$ | $\text{ConvT}_{512} \mapsto \text{BN} \mapsto \text{Relu}$ |
|  |  | $\text{ConvT}_{256} \mapsto \text{BN} \mapsto \text{Relu}$ | $\text{ConvT}_{256} \mapsto \text{BN} \mapsto \text{Relu}$ |
|  |  | $\text{ConvT}_3$ | $\text{ConvT}_3$ |

Table A.1: Model architectures for the different data sets tested. $\text{FC}_n$ and $\text{Conv}_n$ represent, respectively, fully connected and convolutional layers with $n$ output/filters. We apply a component-wise normalisation of the control components which proved crucial for good performance of the model. The dynamic is run on the time segment [0,10] which empirically yields good results.

## C.4 AUToN-CODE ARCHITECTURES

Adapting the models used by Tolstikhin et al. (2017) and Ghosh et al. (2020), we use a latent space dimension of 25 for MNIST, 128 for CIFAR-10 and 64 for CelebA. All convolutions and transposed convolutions have a filter size of 4×4 for MNIST and CIFAR-10 and 5×5 for CELEBA. We apply batch normalization to all layers. They all have a stride of size 2 except for the last convolutional layer in the decoder. We use Relu non-linear activation and batch normalisation at the end of every convolution filter. Official train and test splits are used for the three datasets. For training, we use a mini-batch size of 64 in MNIST and CIFAR and 16 for CelebA in AutoN-CODE. (64 for control models.) All models are trained for a maximum of 50 epochs on MNIST and CIFAR and 40 epochs on CelebA. We make no additional change to the decoder. We train the parameters of the encoder and decoder module for minimizing the mean-squared error (MSE) on CIFAR-10 and CelebA (Liu et al., 2015) or alternatively the Kullback-Leibler divergence between the data distribution and the output of the decoder for MNIST (formulas in Appendix.) Gradient descent is performed for 50 epochs with the Adam optimizer (Kingma & Ba, 2014) with learning rate $\lambda = 1e^{-3}$ reduced by half every time the loss plateaus. All experiments are run on a single GPU GeForce Titan X with 12 GB of memory.

## C.5 VISUALIZATION OF LATENT CODE DYNAMICAL EVOLUTION

## C.6 EXPLORING SAMPLING DISTRIBUTIONS

For random sampling, we train the VAE with a $\mathcal{N}(0, I)$ prior. For the deterministic models, samples are drawn from a mixture of multivariate gaussian distributions fit using the testing set embeddings. The distribution is obtained through expectation-maximization Dempster et al. (1977) with one sin-

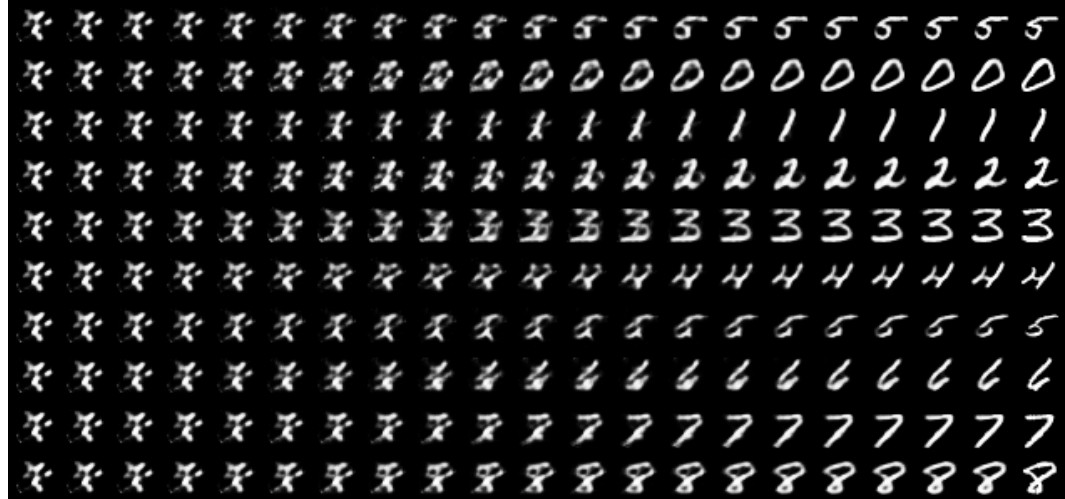

Figure A.4: Reconstructions of the image along the controlled orbits of AutoN-CODE for MNIST. The temporal dimension reads from left to right. Last column: ground truth image

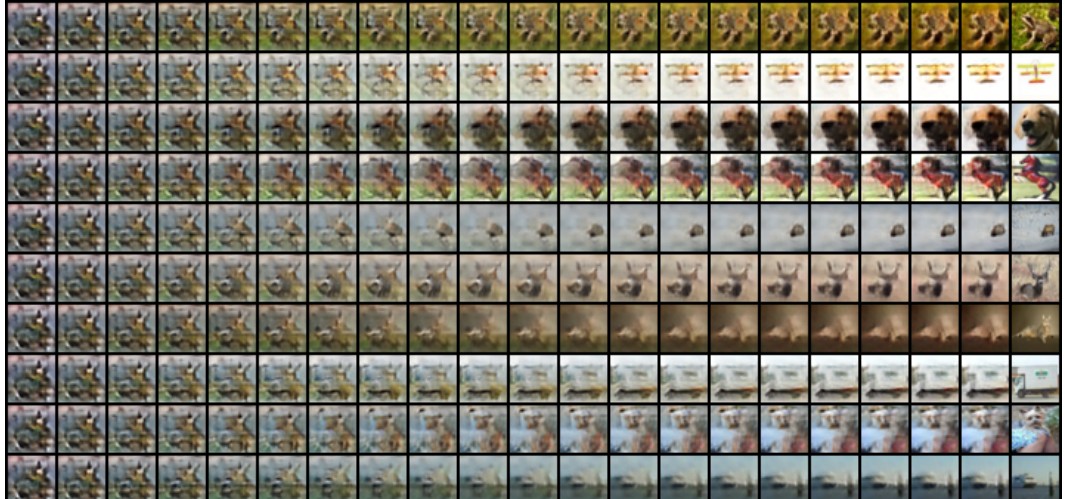

Figure A.5: Reconstructions of the image along the controlled orbits of AutoN-CODE for CIFAR-10. The temporal dimension reads from left to right. Last column: ground truth image.

gle k-means initialization, tolerance $1e^{-3}$ and run for at most 100 iterations. We compute the FID using 10k generated samples evaluated against the whole test set for all FID evaluations, using the standard 2048-dimensional final feature vector of an Inception V3 following Heusel et al. (2017) implementation.

The results show that 10K components (matching the number of data points in the test sets) naturally overfits on the data distribution. Generated images display marginal changes compared to test images. However, 1000 components does not, showing that our AutoN-CODE sampling strategy mediates a trade-off between sample quality and generalization of images. We alternatively tested non-parametric kernel density estimation with varying kernel variance to replace our initial sampling strategy. We report similar results to the gaussian mixture experiment with an overall lower FID of AutoN-CODE for small variance kernels. As the fitting distribution becomes very rough ($\sigma \approx 5$), the generated image quality is highly deteriorated.(see Figure A6).

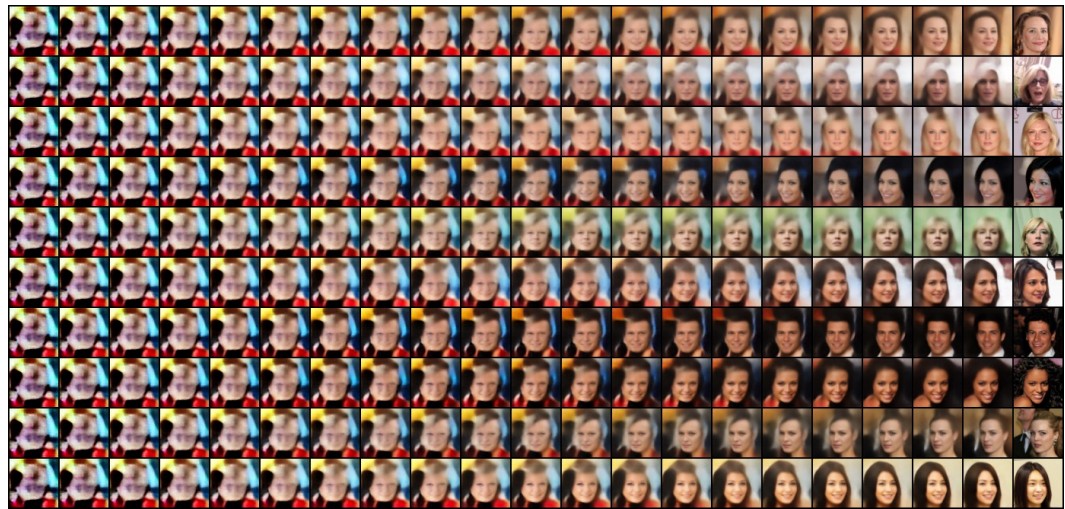

Figure A.6: Reconstructions of the image along the controlled orbits of AutoN-CODE for CelebA. The temporal dimension reads from left to right. Last column: ground truth image

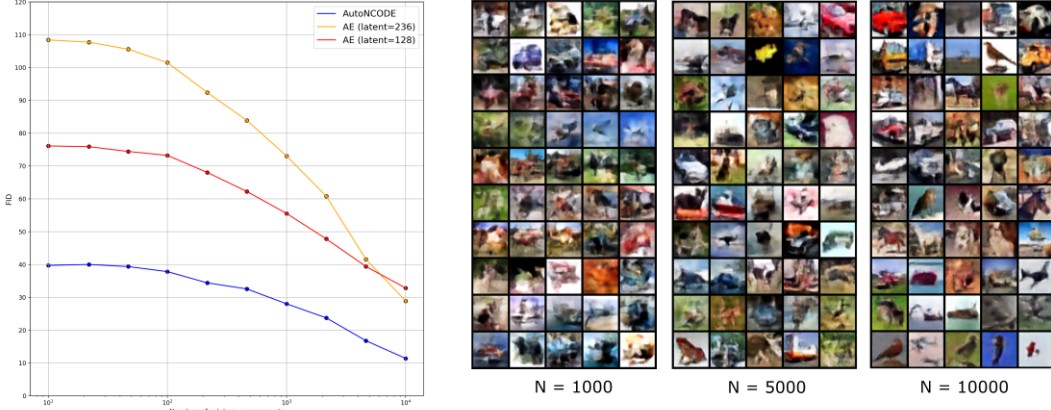

Figure A.7: **Left:** Evolution of FID as a function of the number of components in the Gaussian mixture used to fit the latent distribution. We also tested a vanilla autoencoder with twice the capacity (236-dimensional), exactly matching the number of parameters of our model. Interestingly, this model performs worst in a low component regime. We interpret this a as a manifestation of the "curse of dimensionality", as the latent example population becomes less dense in this augmented space, making the naive gaussian mixture fitting less accurate for fitting the latent distribution. **Right:** Samples of generated images for different number of mixture components. The shape and details becomes clearer with increasing components.

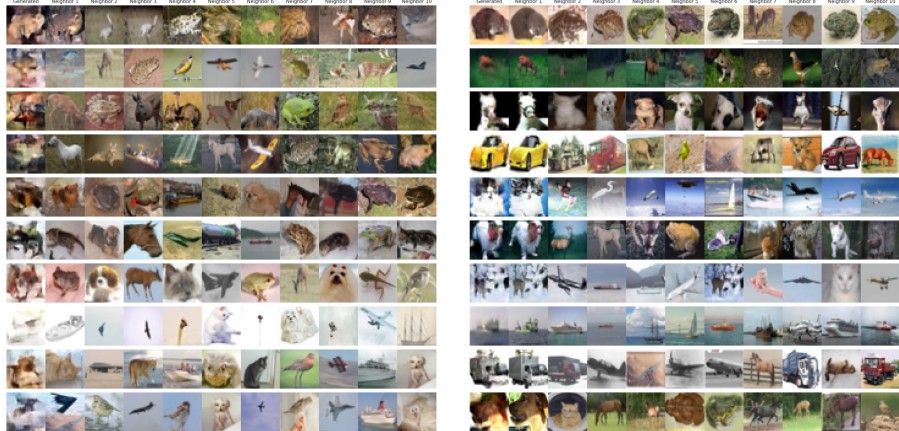

Figure A.8: Nearest neighbors search of random samples from the gaussian mixture with (**Left**) 1000 components (**Left**) and (**Right**) 10K components in the testing set of CIFAR-10. The sampled latents show overfiting on the test set in the highest case.

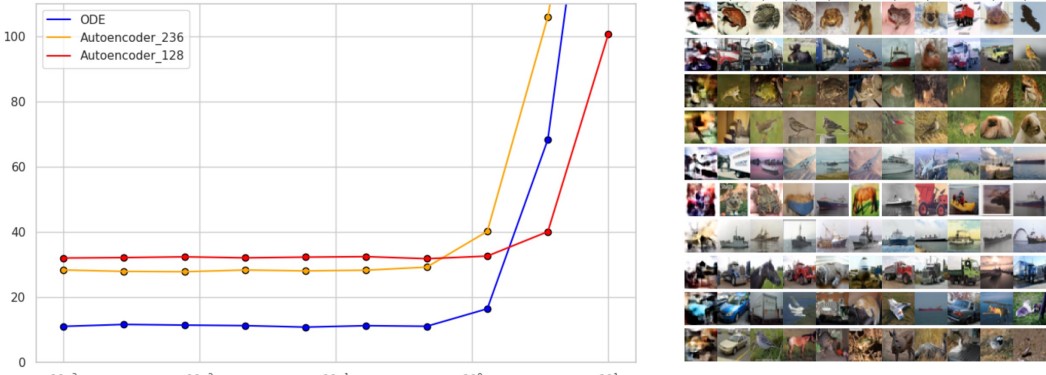

Figure A.9: (**Left**) Evolution of FID between test set and generated samples as a function of the gaussian kernel variance used to perform kernel estimation of the the latent code distribution. (**Right**) Nearest neighbor search of random samples from the latent distribution fit by gaussian kernel estimation with variance $\sigma = 2$ in the testing set.

## C.7 AUTON-CODE LATENT CLUSTERING

We further investigate the representations learned by our model by measuring its clustering accuracy against several other techniques in Fig. A.10. Similarly to the image generation experiment, we consider the final state of the dynamical system as the latent representation that we cluster using a 10 component gaussian mixture with different initializations. We did not perform any supplementary fine-tuning or data augmentation to train our model, but we also tested a version with further dimensionality reduction using t-SNE (Maaten & Hinton, 2008). The results, although inferior to recent deep clustering techniques, show better clustering accuracy than other autoencoding models, suggesting a different organization of the latent code compared to the vanilla linear projection of the autoencoder.

| Clustering accuracy | | |
| --- | --- | --- |
| Model | CIFAR-10 | MNIST |
| K-means (Lloyd, 1982) | 22.9 | 57.2 |
| Spectral clustering (Shi & Malik, 2000) | 24.7 | 69.6 |
| Variational Bayes AE† (Kingma & Welling, 2013) | 29.1 | 83.2 |
| Sparse AE (Ng, 2011) | 29.7 | 82.7 |
| Denoising AE (Vincent et al., 2010) | 29.7 | 83.2 |
| AE (GMM)† | 31.4 | 81.2 |
| GAN (2015) (Radford et al., 2015) | 31.5 | 82.8 |
| DeepCluster (Caron et al., 2018) | 37.4 | 65.6 |
| DAC (Chang et al., 2017) | 52.2 | 97.8 |
| **IIC** (Ji et al., 2019) | **61.7** | **99.2** |
| AutoeN-CODE (GMM)† | 33.31 | 86.02 |
| AutoeN-CODE (t-SNE + GMM)† | 27.00 | 97.26 |

Figure A.10: Unsupervised image clustering accuracy on CIFAR-10 and MNIST against recent models. Results obtained with the authors original code are noted with †.

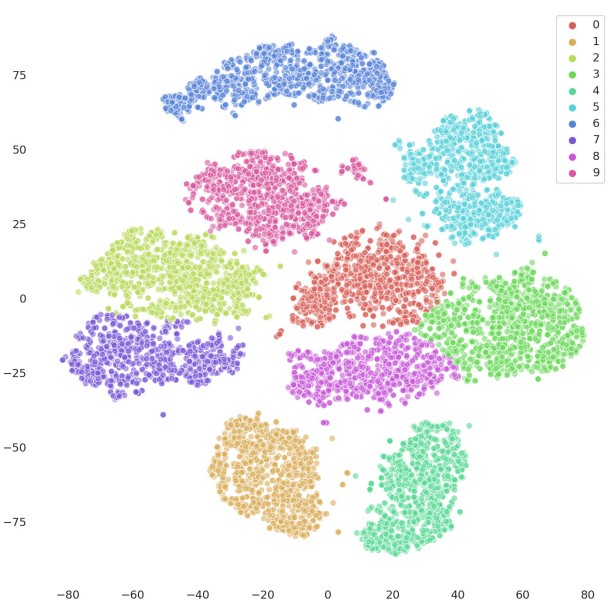

Figure A.11: tSNE embeddings of the latent code at $t = t_1$ for MNIST test set colored with a 10 component gaussian mixture model.

## C.8    LATENT CODE INTERPOLATION

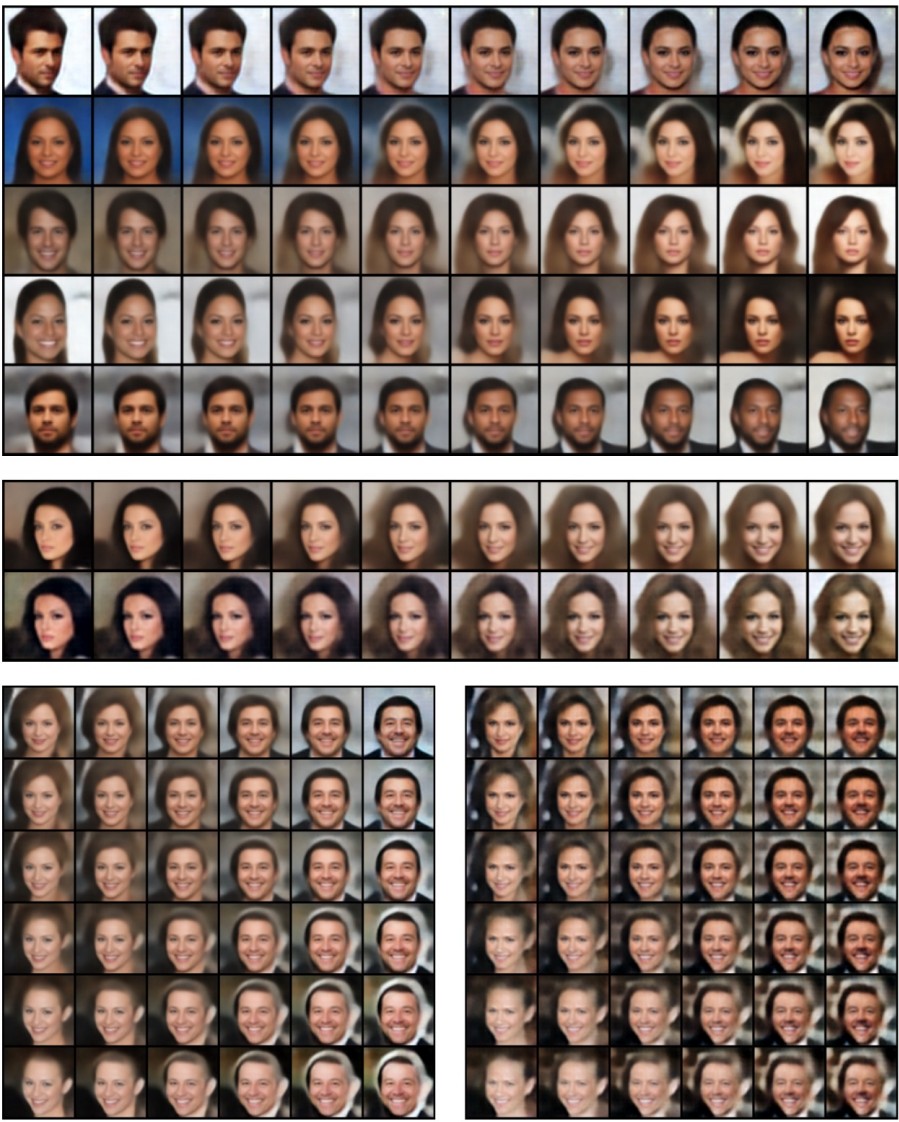

Figure A.12: Interpolation: We further explore the latent space of AutoN-CODE by interpolating between the latent vectors of the **CelebA** test set. (**Upper panel**) Linear interpolation between random samples reconstructed with AutoN-CODE. (**Middle panel**) Interpolation comparison between AutoN-CODE and a vanilla autoencoder for a single pair of vectors. (**Lower panel**) 2d interpolation with three anchor reconstructions corresponding to the three corners of the square (A:up-left,B:up-right and C:down-left. Left square corresponds to AutoN-CODE reconstructions and right to a vanilla autoencoder.

