# OpenReview forum: "Go with the flow: Adaptive control for Neural ODEs"
_ICLR.cc/2021/Conference — ICLR 2021 Poster_

### Official Review · AnonReviewer1 · 2020-10-22
**Improving neural ODEs with time varying weights and data dependent vector fields**

**Rating:** 7
**Confidence:** 4

**Review:**


**Paper summary**
The paper proposes a new class of neural ODE based models, called neurally-controlled ODEs (N-CODE). Instead of directly learning the weights of the neural network parameterizing the vector field f of the ODE, the paper instead proposes to learn a controller that takes as input the initial state and outputs the initial weights of f and optionally also updates the weights of f as the ODE is solved. This allows the model to parameterize a family of (potentially time varying) vector fields instead of a single vector field. The authors go on to show that their model can be trained using an augmented adjoint method (similar to the adjoint method used in the original neural ODE paper).

The authors test the performance of the proposed model across a variety of tasks. First, they test their model on some of the toy problems proposed in the augmented neural ODEs paper. They show that their model can successfully learn mappings to represent the 1D reflection and 2D concentric annuli functions. For example, in the case where the model only learns a controller for the initial state, the model learns to map +1 and -1 to different vector fields, effectively allowing the model trajectories to cross. Similarly, in the case where the vector field is additionally controlled as time evolves, the model also allows states to cross and represent both functions. Second, the authors evaluate their model on a supervised learning task, showing that the proposed method improves performance on both MNIST and CIFAR10.

Finally, the authors build an interesting new form of autoencoder based on their method. The autoencoder takes as input an image and outputs the initial weights of the vector field parameterizing an ODE. The authors then allow a fixed initial state to evolve in this vector field. The final state is then used to decode the model into the reconstructed image. Therefore, the latent information about the image is contained in the weights of the vector field as opposed to a latent vector as is usually the case. The authors evaluate their method on both CIFAR10 and CelebA using the methods from the deterministic autoencoder paper to endow their model with a latent space they can sample from. The results on both datasets are generally good.


**Positives**
- The paper is clearly written and the model is well explained. The figures are also nice.
- The autoencoder experiments are interesting and novel.
- The model performs well across a wide variety of tasks from toy problems to image classification and autoencoder like unsupervised learning.
- The motivation for the paper is clear and fairly important: alleviating some of the representational weaknesses of neural ODEs and generally improving and extending neural ODE models by using time varying weights.

**Negatives**
- The paper is closely related to other works that have also focused on making the weights of neural ODEs time dependent. Data controlled neural ODEs from the dissecting neural ODEs paper are, as far as I can tell, exactly the same as the open loop model proposed in this paper. Further, the dissecting neural ODEs paper also proposes using time varying weights (as do some other papers), although not in exactly the same way as this paper. As such the novelty of this paper is quite limited (although the autoencoder experiments are, to the best of my knowledge novel). It would be good to have a more thorough discussion of the differences between these models to better understand where the novelty/contribution comes in.
- The closed loop model (which is the main innovation) is thoroughly discussed in the paper. However, it seems like most experiments are actually performed with the open loop model (which is very similar to other models already proposed in the literature). The image classification experiments and the autoencoder experiments both use (as far as I can tell), the open loop model. The only place where the authors seem to use the closed loop model is in the toy experiments, where the model performs worse than the simpler open loop model. This does put into question whether the full closed loop model is actually useful in practice.
- There is no evaluation of the number of functions evaluations in the paper (as far as I can see). According to the abstract, the model trains faster so it seems important to include this information in the paper.
- The authors mention in the abstract that the model achieves state of the art reconstruction on CIFAR10. However, reconstruction doesn’t really make sense as a task (since the identity function would be optimal in this case). It is interesting if the latent space learned is very low dimensional (i.e. the model compresses the dataset well), however there seems to be no discussion of this. So I feel like this claim is questionable and should be removed.

**Recommendation**
While the paper is well written and has thorough experiments, the proposed model is very closely related to several models that have already been proposed in the literature. Further, the full model seems to not be used in most of the experiments and appears to perform worse than the simplest version of the model. The autoencoder experiments however are quite compelling and novel and would likely be of interest to the ICLR community. I therefore recommend a weak accept.

**Typos and small comments**
- In the first sentence of the paper the word “opens” seems like it shouldn’t be there
- The word “such” is repeated on bottom of page 2
- There are no labels on the Figure A.5 axes

---

> ### Author Response · Authors · 2020-11-17
> **Response to AnonReviewer1**
>
> The reviewer’s remarks, especially those concerning the distinction between our system and related models and the value of the closed-loop model, are well-taken. We have addressed these remarks in the revised manuscript and we believe the resulting theoretical and empirical description of our model is much strengthened as a result. In detail:
>
>  - We have included a new passage in Sec. 3.1 in the  “Open-loop Control” paragraph comparing NCODE to concurrent work by Massaroli and others. In this section, we state that appending data to the dynamic variable is one form of data control similar in spirit to the time-concatenation trick used to make NODEs non-autonomous systems, but that there are many forms of dynamics that we can explore, notably closed-loop control. Our formulation, which, unlike earlier work, makes an architectural distinction between controller and dynamics, makes generalization to this other type of control easy.
>
> - Consequently, and in light of your remark on the lack of experimental results for the closed-loop system, we have added a new experiment in which the special benefits of this formulation are more evident.
>
> - This experiment (new section 4.3) relies on our intuition that the closed-loop formulation is especially useful on problems involving non-stationary stimuli in which processing must occur “on-the-fly”. This simulation shows how closed-loop NCODE vastly outperforms competing models on a pattern memorization task in which systems view several high-dimensional bit vectors and must then reconstruct a degraded version of one of them from memory. This new experiment shows how closed-loop control not only beats other continuous-time models, but also the open-loop NCODE formulation. It further empirically demonstrates the value of our system over the related open-loop formalism of Massaroli, in which controller and equation of motion are not architecturally distinct.
>
> - Finally, we agree with reviewer 1 on the image reconstruction claim. We changed the formulation in the abstract and body, now emphasizing the improvement that latent flows induce when added to a vanilla autoencoder.

---

> > ### Comment · AnonReviewer1 · 2020-11-20
> > **Thank you for addressing concerns**
> >
> > I'd like to thank the authors for addressing the two main concerns I had, namely:
> > - There was not a clear discussion of the distinction between the contributions of this work and those of closely related papers
> > - There were no experiments showing the usefulness of the closed-loop formulation of the model (which is the main contribution of the paper)
> >
> > The updated section 3.1 in the paper addresses the first concern well. I appreciate the updated discussion and clear distinction this model has with the ones proposed in the dissecting neural ODEs paper. The second concern is well addressed by the pattern memorization experiments which are very cool and show a compelling case for the use of the closed-loop model.
> >
> > I also appreciate the updated formulation of the image reconstruction claim for the autoencoder task. I think that section reads much better now.
> >
> > Because of these improvements I have raised my score from 6 to 7 and I think this paper deserves to be accepted at ICLR.

---

### Official Review · AnonReviewer2 · 2020-10-26
**Recommendation to Accept**

**Rating:** 8
**Confidence:** 4

**Review:**

Comment: Summary: This paper presents a technique for more expressive neural ordinary differential equations (NODE) flows. Instead of learning a fixed set of parameters $\theta$ that governs the ODE dynamic, the proposed approach learns dynamic parameters that evolve over time. The authors propose two variants of the methods: $\textit{open-loop}$ and $\textit{closed-loop}$. The former only maps the initial observation as the controller whereas $\theta$ in the later model follows another NODE $g$. Model performance is demonstrated on several tasks. The model is shown to solve the well-known "crossing curves" problem, on which NODE fails. Also, it's demonstrated that the presented technique yields sharper images compared to VAEs and also better at classification and interpolation.

Overall Score: I recommend an accept.
- First of all, the presented N-CODE method solves one of the most crucial limitations of NODEs in a principled manner. The model is also shown to outperform the vanilla Augmented NODE method.
- The results are very impressive. Both the tables and generated images/interpolations are of high-quality, especially given VAEs don't excel at this task.
- The method has certain overlaps with the control theory and maximum principle, deep generative models, and neural nets with adaptive weights. So I believe such connections would open new research avenues.

Cons: I would be happy if the below are addressed:
- Did you investigate how the model performance changes as f grows? I speculate that learning the parameters of a neural network via another neural network(s) is a very challenging problem, and would like to see this is verified or not. Also, I cannot see the architecture of f in your experiments (looking at Table A.1).
- Connection with control theory can be made clear. As such, there is very little reference to Pontryagin's maximum principle and the link is not visible (at least to me).
- Did you test vanilla NODE on experiments 5.3 and 5.4? The virtue of N-CODE is obvious on the toy problem (as expected) and somewhat significant on the classification task. I'm wondering when NODE is latent (as in 5.3-5.4), is the improvement significant?

Additional comment: Is Figure-4 caption correct?

---

> ### Author Response · Authors · 2020-11-17
> **Response to AnonReviewer2**
>
> We thank the reviewer for the thoughtful remarks which we think will help strengthen the paper. In response to these remarks:
>
>  - We have included Included additional remarks (Sec. 3.2) about the bearing of control theory on our system. In particular, we have clarified the link between the maximum principle and our adjoint formulation. We emphasized that we propose to solve an augmented adjoint system that encompasses the weights as control variables of the system. When defining meta-parameters affecting the dynamics of \theta, this formulation subsequently lift the optimization problem into a functional space.
>
> - We will add in an upcoming update to section C.2., an experiment  discussion in the supplement in which we test different forms of control for the function f in the image classification experiment, where the number of controlled parameters in f is varied. The design of this experiment can already be read in section C.2.
>
> - We have reframed the unsupervised learning section to be specifically about the improvement in image generation caused by the addition of latent flows to a vanilla autoencoder. We thank the reviewer for the suggestion to explore the effect of NODE latent transformations, as we haven’t tested this configuration. As our interest is not the absolute performance of our system compared to various other models (now redacted), we believe that comparisons to other continuous-time models can be left for future specific work.

---

> > ### Comment · AnonReviewer2 · 2020-11-21
> > **Nice Response**
> >
> > Thanks to the authors for their response! The paper reads better now and the connection with previous work is more clear. I would still be happier to see comparisons with the vanilla methods but the experiments overall are convincing.

---

### Official Review · AnonReviewer3 · 2020-10-28
**Needs better motivation and results, and clarity on relation to previous work**

**Rating:** 7
**Confidence:** 5

**Review:**

Update:

I am happy with the authors' rebuttal and have increased my rating accordingly.

--------------------------------

Summary

The paper proposes to predict the required weights of a Neural ODE function from the input data. The paper has two parts - supervised classification, and unsupervised image reconstruction and generation. In the supervised part, has good visualizations of the decision boundaries induced. The unsupervised part shows good comparisons with previous methods, and good visualizations of the reconstructed and generated images.

Strengths

The paper is well organized. The images and figures are explained well. The graphs showing the trajectories of the Neural CODE are great for visualization. The paper provides enough details about training its networks.

Comments

The paper is split into two parts - supervised classification, and unsupervised modeling. The supervised classification part is highly related to the “Data Controlled Neural ODEs” section in Massaroli et al (2020b), a paper that has been cited in the related work section but not addressed in the main content. In fact, the problems tackled (-x, concentric annuli) and the results are highly related to those of Massaroli et al (2020b). This issue needs to be addressed sufficiently.

The unsupervised section needs a lot more work. The experiments and tables can be described more effectively. For example, it would be preferable to clearly explain which row section 5.3 refers to in Figure 7, and which row corresponds to section 5.4.

The fact that the replacement of a linear layer with a Neural CODE improves image reconstruction quality should mean that all layers in the encoder can be replaced to give better encoding, taking care of dimensionality (such as in Normalizing Flows). In fact, generative modelling papers such as FFJORD, or Normalizing Flows, do replace all layers with a Neural ODE and map to a latent space of noise. However, this paper uses a typical neural network for image reconstruction and generation. This perhaps means that the majority of heavy-lifting is done by the decoder, hence the Neural CODE is more amenable to warping the latent space suitable for the decoder. For image generation as well, the latent space has been designed so that the decoder can produce nice images, this is not necessarily a win for the Neural CODE.

To make the case for Neural CODE, especially for images, higher resolution images need to be tackled, since it is in higher dimensions that the success of the advancements in the methods listed in Figure 7 lies.

Massaroli et al. (2020b) : Dissecting Neural ODEs; NeurIPS 2020;  https://arxiv.org/abs/2002.08071
Grathwohl et al. : FFJORD : FFJORD: Free-form Continuous Dynamics for Scalable Reversible Generative Models

Minor

Many variables needs to be bold at multiple places.
Figure 4 needs to say “Third” for the closed loop figure.
Zhang et al., 2019b is cited in related work, but needs to be cited along with the respective experiment for Figure 5.

---

> ### Author Response · Authors · 2020-11-17
> **Response to AnonReviewer3**
>
> We thank the reviewer for calling our attention to the works of Dupont and Massaroli, the latter of which was concurrently published at the last NeurIPS after a version of this paper was submitted. Nevertheless, and we believe that a more explicit comparison between our work and this other material will strengthen the manuscript. In addition to discussing this other work in greater detail, we have also:
>
>  - Added a brief section ( in Sec. 3.1., “Open-loop Control”) comparing NCODE to the data-controlled NODEs of Massaroli et al. We note how two differ primarily by our notational decision to separate controller and dynamics into separate functions. We argue further, however, that this notational decision arises from our system’s generalizability to non-stationary control parameters, which we note next.
>
>  - Included a new experiment (Sec. 4.3) involving non-stationary input in which closed-loop NCODE performs favorably compared to the data-controlled NODEs of Massaroli et al., demonstrating the difference between our method and that of the other authors as well as the special value of our control formulation for non-stationary environments.
>
>  - We also agree with the reviewer that the relative contributions of the decoder and latent NCODE representations to performance in the unsupervised setting were not explained clearly enough in the original manuscript. We have adjusted the manuscript to emphasize that that the encoder/decoder architectures in both the vanilla autoencoder and the NCODE version are identical so that, as the reviewer suggests, “[N]CODE is more amenable to warping the latent space suitable for the decoder” than having no warping at all. We have also added unsupervised clustering results on CIFAR-10 and MNIST for the AutoNCODE latent space which suggest that a different organization of the latent space from a vanilla autoencoder (Sec. C.7).
>
>  - The reviewer’s suggestion that the model should be applied to higher resolution images is well-taken and we hope to perform this experiment in the future.  In accordance with the reviewer’s suggestion, we have reframed the image generation section of the paper to argue that NCODE can be used to boost the performance of a vanilla autoencoder architecture, producing competitive results on a well-worn data set, rather than act as a state-of-the-art supervised generative method in its own right.
>
> - Finally, we have corrected the typographical errors and clarified the presented results as pointed out by the reviewer.

---

### Official Review · AnonReviewer4 · 2020-10-31
**The flow is strong with you N-CODE**

**Rating:** 7
**Confidence:** 2

**Review:**

The paper introduces a novel approach N-CODE, based on Neural Ordinary Differential Equations (NODE), that increases the expressivity of  continuous-time neural nets by using approaches from Control theory. N-CODE, in contrast to NODE, can learn a family of vector fields and is therefore able to flexibly adjust the flow for every data point. Authors provide theoretical evidence and compute several simulations with different problems (including Reflection, concentric annuli, supervised image classification and image auto encoding) that confirm that the proposed method greatly improves or outperforms state-of-the-art methods.

The paper is well written and clear. The proposed approach is original, and the results suggest that the performance of the approach leads to significant improvements compared to the state-of-the-art.

The performance of the method was assessed by applying N-CODE to different problems. While evidence in terms of performance is presented, no theoretical or empirical evidence is presented that confirms that the training speed is significantly lower compared to NODE. Also, while performance seems generally higher, the limitations of the proposed method are not clear. Better performance in less time is a strong claim, that is not fully supported by the results.

In summary, however, this is a very  interesting approach with potential.

---

> ### Author Response · Authors · 2020-11-17
> **Response to AnonReviewer4**
>
> We thank the reviewer for the kind remarks and agree that our discussion on NCODE learning efficiency and limitations should be strengthened. In response to this suggestion, we have adjusted the manuscript to clarify the particularities of training NCODE and included new empirical evidence for improved convergence in a new experiment. In particular:
>
>  - We included a new experiment (Sec. 4.3) using closed-loop control in which convergence is accelerated over competing models by several orders of magnitude.
>
>  - We will soon add in another update to section C.2 (in appendix), and comment training curves (loss, accuracy, NFE) for the image classification experiments.

---

> > ### Comment · AnonReviewer4 · 2020-11-23
> > **Follow-up**
> >
> > Thank you for the clarifications and adding the n-bit pattern memorization task and the results that show that the proposed method converges quicker for this task than baseline methods. Good paper.

---

### Author Response · Authors · 2020-11-17
**Collective response to reviewers**

We are very grateful to all four reviewers for the time taken in assessing our work and for the sensible  and overall encouraging feedback, that we think will help improve the submission.

We have added new favorable results leveraging the full proposed NCODE framework in a few-shot memorization task and clarified relation of our work to the relevant litterature. We believe that this has significantly improved the paper and we sincerely hope that AnonReviewer3 would consider increasing his rating.

We will update a definitive version before the end of the revision period in which we will provide additional details in relation with reviewers comments.

---

### Decision · Program_Chairs · 2021-01-07
**Final Decision**

**Decision:**

Accept (Poster)

**Comment:**

This paper introduces a few variants of neural ODE architectures to improve their expressivity.  The motivation and method make sense, but are fairly incremental.  The tasks are also fairly low dimensional and as one reviewer pointed out, reconstruction isn't a good benchmark task.

However, the paper seems well-executed, and the rebuttals answered the expert rewiewers' concerns.